# Femoral anterior condyle height decreases as the distal anteroposterior size increases in total knee arthroplasty: A comparative study

Bo Yang[1,2,ø,¤], Fu-zhen Yuan[1,ø], Hai-jun Wang[1], Xi Gong[1], Yan-hai Chang[2]*, Jia-Kuo Yu[1,3]*

1 Sports Medicine Department, Beijing Key Laboratory of Sports Injuries, Peking University Third Hospital, Beijing, China, 2 Department of Orthopaedics, Shaanxi Provincial People's Hospital, The Third Affiliated hospital of Xi'an jiaotong University, Xi'an, China, 3 Orthopaedic Sports Medicine Center, Beijing Tsinghua Changgung Hospital, Affiliated Hospital of Tsinghua University, Changping District, Beijing, China

ø These authors contributed equally to this work.
¤ Current address: Department of Orthopaedics, Shaanxi Provincial People's Hospital, The Third Affiliated hospital of Xi'an jiaotong University, Xi'an, China
* yujiakuo@tsinghua.edu.cn (JKY); changyanhai123@126.com (YHC)

**Data Availability Statement:** All relevant data are within the manuscript and its Supporting Information files.

## Abstract

### Purpose

The anterior flange height of the current femoral component increases with an increasing distal femoral anteroposterior dimension. During total knee arthroplasty (TKA), we have observed that a large femur may have a thinner anterior condyle, whereas a small femur may have a thicker anterior condyle. The first purpose of this study was to examine whether the femoral anterior condyle height decreases as the distal femoral anteroposterior size increases and whether gender differences exist in anterior condyle height.

### Methods

A total of 1218 knees undergoing TKA intraoperative and computed tomography scans from 303 healthy knees were used to measure the anterior lateral condylar height (ALCH), anterior medial condylar height (AMCH), and the lateral anteroposterior (LAP) and medial anteroposterior (MAP) dimensions of distal femurs. The LAP and MAP measurements were used for adjustments to determine whether gender differences exist in anterior condyle heights. Linear regression analysis was performed to determine correlations between ALCH and LAP or between AMCH and MAP.

### Results

There were significant differences between males and females in ALCH in both the CT and TKA groups and AMCH in the CT group (all P<0.01). After adjusting for LAP and MAP, there were significant gender differences in the lateral and medial condylar heights in both groups (P<0.01). There were significant negative correlations between ALCH and LAP values and between AMCH and MAP values in both CT and TKA measurements, with the LAP and MAP values increasing as ALCH and AMCH decreased.

**Funding:** We declare that this study was approved by institute review board of author' hospital (No: IRB00006761-2011073). This research was funded by Instrument Research Project of the National Natural Science Foundation (No. 81327001), which took part in study design, data collection and analysis, preparation and decision to publish the manuscript; Science and Technology Program of Shaanxi Province (No.2022SF-049), which took part in preparation and revised the manuscript; Capital's Funds for Health Improvement and Research (No.2022-1-4091) and National Natural Science Foundation-Beijing Natural Science Foundation Regional Innovation Union Development Fund (No.U22A20283).

**Competing interests:** The authors have declared that no competing interests exist.

## Conclusions

The results demonstrate that femoral anterior condylar height decreased with increasing anteroposterior dimension in both the medial and lateral condyle. In addition, this study also showed that anterior condylar heights are highly variable, with gender differences. The data may provide an important reference for designing femoral anterior flange thickness to precisely match the natural anterior condylar anatomy.

## Introduction

Although total knee arthroplasty (TKA) is successfully used to treat advanced osteoarthritis of the knee [1], patellofemoral complications are still a main cause of dissatisfaction with the outcome after TKA [2, 3]. In addition to surgical technique (implant positioning, soft-tissue balancing, etc.), patellofemoral component design (patellar component design, femoral component design, etc.) affects the patellofemoral biomechanics associated with these complications [4, 5]. Many studies have focused on the patellofemoral anatomic morphology for properly designing knee components [6, 7]. The anterior flange thickness of the femoral component strongly influences the biomechanics of the patellofemoral joint [8]. If the anterior flange thickness of the femoral component is larger than the resected anterior condyle, there is a risk of overstuffing, anterior knee pain, and limiting the postoperative knee range of motion; if it is smaller than the resected anterior condyle, there is a risk of decreased quadriceps moment arm, resulting in a functional disadvantage [6, 9].

One of the design details considered to be important is restoration of the natural anterior condylar height. Several studies have reported that the anterior condylar height positively correlates with the length of the femur or femoral size [10, 11]. Traditionally, the anterior flange thickness of currently used femoral components increases as the component size increases [12]. However, while performing TKA, we have observed that a large femur may have a thinner anterior condyle, and a small femur may have a thicker anterior condyle. This raises questions about whether the anterior condylar height increase as the femoral size increases and whether the anterior flange currently used in femoral components mismatch the natural anterior condyle anatomy.

The anteroposterior (AP) dimension was used to select an implant of appropriate size during TKA. No study has yet analyzed the relationship between anterior condyle height and distal femoral AP size. The aim of the present study was to examine the relationship between anterior condyle height and distal femoral AP size to provide data for designing size-matched anterior flanges for femoral components. We hypothesize that femoral anterior condyle height decreases as the distal femoral AP size increases.

## Materials and methods

This study was approved by the institutional review board of Peking University Third Hospital. Prior to the start of the study, all participants provided informed verbal consent.

### Method for intraoperative measurement

Between August 2011 and September 2015, a total of 1218 knees in 222 male and 996 female primary OA patients undergoing TKA were evaluated in this study. The mean patient age was $67.3 \pm 7.8$ years for males and $65.6 \pm 6.9$ years for females. Patients were excluded if they had a

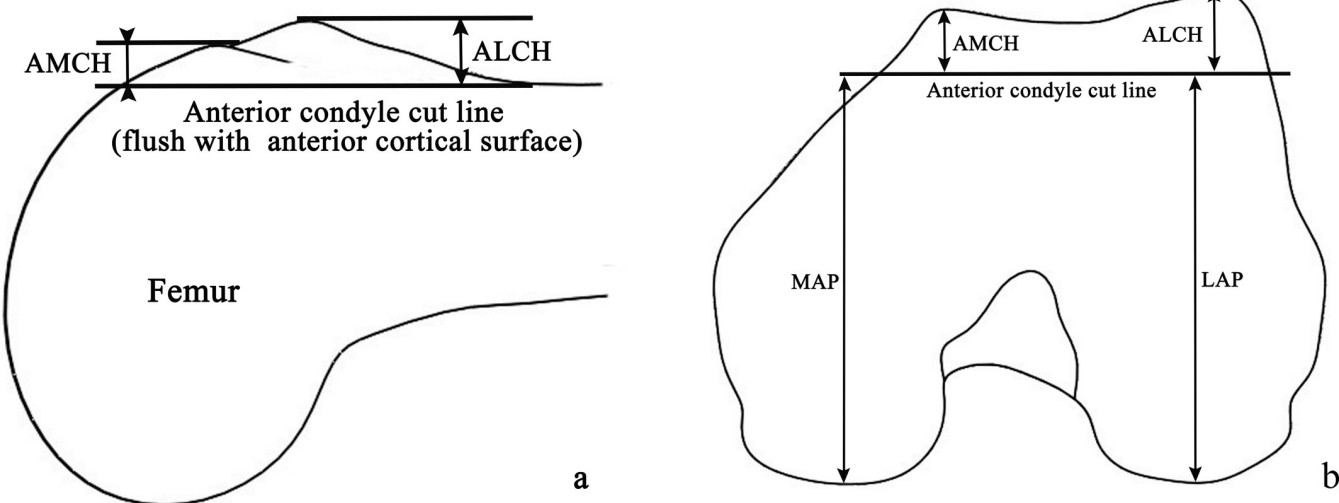

**Fig 1. Diagram of cuts and measurements in the distal femur.** (a) Anterior condylar cut line, which is flush with the anterior cortical surface of the distal femoral shaft; (b) Measurements of the distal femur. ALCH: anterior lateral condylar height; AMCH: anterior medial condylar height; LAP: lateral anterior-posterior; MAP: medial anterior-posterior.

history of femur fracture or congenital anomaly or if the bone loss or degradation was so serious that augmentation was required or if the knee had a varus or valgus deformity of >15˚. All TKA procedures were performed by the senior surgeon (Yu). The NexGen knee prosthesis system (Zimmer, Warsaw, IN, USA) was used for all patients.

Intramedullary femoral alignment was used to make a 10 mm distal femoral resection at 5–7˚ valgus to the anatomic axis. The femoral external rotation 3˚ relative to the posterior condylar axis or paralleling the transepicondylar axis or perpendicular to Whitesides' line for the distal femoral cut. Anterior referencing instrumentation was used to cut the anterior condyle. The anterior condylar heights were measured intraoperatively only for patients whose anterior condylar cut line flushed with the anterior cortical surface of the distal femoral shaft (Fig 1A). Patients whose anterior condylar cut lines were above or below the anterior cortical surface were excluded. All visible osteophytes were removed before measurements were taken, and all measurements were taken twice: once by the surgeon and the second by his assistant to reduce random error. The anterior lateral condylar height (ALCH) and anterior medial condylar height (AMCH) are the maximum thickness of the cut-down at the anterior lateral and medial condyle, respectively. The femoral lateral anterior-posterior (LAP) and medial anterior-posterior (MAP) dimensions were defined as the distance between the lowest lateral and medial condyle, respectively, and the anterior condyle cut surface (Fig 1B).

## Measurement on computed tomography

Between August 2011 and December 2012, a total of 152 Chinese males and 151 females were recruited at the time of computed tomography (CT) of the knee after anterior or posterior cruciate ligament reconstruction. The mean age was 32.0 ± 10.9 years for males and 27.9 ± 7.2 years for females. Subjects were excluded if they had a history of femoral fracture or congenital anomaly, if they had diseases that could affect the normal formation of the knee joint, or if the knee had a varus or valgus deformity greater than 10˚.

A CT scan of each knee was performed using a helical CT scanner (120 kVp, 200 mA, Somatom Sensation 16, Siemens Healthcare, Germany). The patient was placed supine with

the knee in a full extended position on the scanner with the patella facing towards the ceiling. The scanning procedure acquired 1-mm CT slices (image size, 512 × 512 pixels). All CT images were burned into discs. The images of the knees were segmented using a region-growing method to construct 3D bony models. To simulate TKA anterior condyle cuts and measure the femoral condyle geometry, a line connecting the medial sulcus of the medial epicondyle and the lateral epicondylar prominence was defined as the surgical transepicondylar axis (STEA) created on each knee model. An orthogonal coordinate system was then established at the lateral end point of the STEA on the surface model. The X-axis was defined as the STEA; the Y-axis was defined as a line parallel with the femur mechanical axis, which is 6° valgus with the anatomical axis (the line along the femoral shaft) in the coronal plane (Fig 2A); and the Z-axis was vertical to the X- and Y-axes, passing through the lateral end point of the STEA (Fig 2B). An anterior condyle flush cut along the anterior femur cortex surface, parallel to the femoral shaft and without anterior femoral notching, was performed on each femur (Fig 2A).

The ALCH and AMCH were defined as the maximum thicknesses of the anterior cut at the lateral and medial condyles, respectively. The femoral LAP and MAP dimensions were defined as the distance between the lowest lateral and medial condyle, respectively, and the anterior condyle cut surface (Fig 2B).

## Statistical analysis

SPSS software 18.0 (SPSS, Chicago, IL) was used for statistical analysis. Dimensions were summarized as means and standard deviations. A Kolmogorov-Smirnov test for normality was performed, and the data were found to be normally distributed. Comparative means between males and females were obtained using the independent t-test. The adjusted P-values were

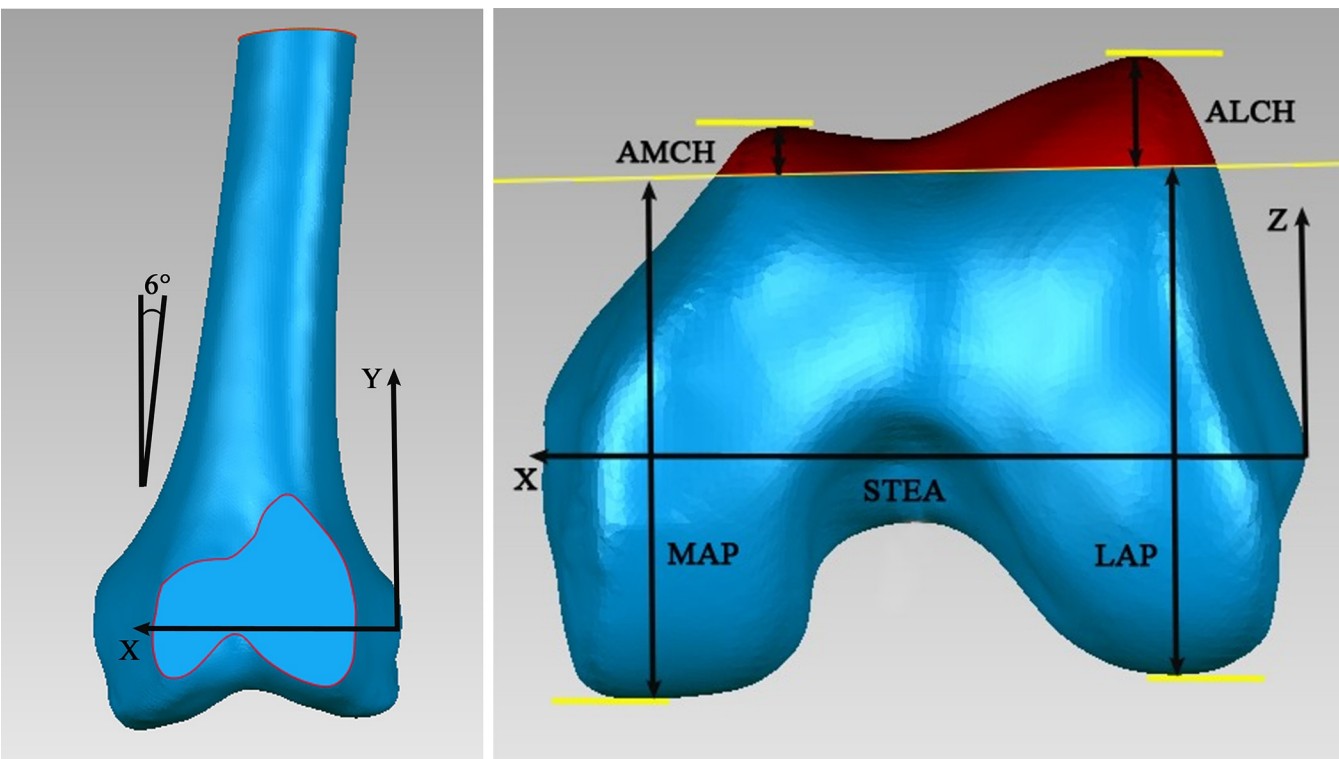

**Fig 2. The anterior femoral condyle cut and measurements.** (a) The anterior femoral condyle cut viewed in the coronal plane. (b) The anterior femoral condyle cut and measurements viewed in the transverse plane.

**Table 1. Anterior lateral condylar dimensions (mm).**

|  | Male |  | Female | P-value | Adjusted for LAP |
|---|---|---|---|---|---|
| CT group | 7.73 ± 1.92 |  | 6.60 ± 1.65 | <0.01 | <0.01 |
| TKA group | 8.36 ± 2.71 |  | 7.46 ± 2.34 | <0.01 | <0.01 |

Values are mean ± standard deviation. LAP: lateral anterior-posterior.

determined by logistic regression. The AMCH was adjusted using the MAP dimension, and ALCH was adjusted using the LAP dimension to determine whether there were gender differences in the anterior condylar height. Linear regression analysis was used to determine correlations for the ALCH and LAP, AMCH and MAP dimensions. The differences were considered significant when P<0.05.

## Results

### Anterior lateral and medial condylar heights

The mean ALCH of males was significantly larger than females in both the CT and TKA groups (P<0.01). The mean ALCH was 7.7 ± 1.9 mm for males and 6.6 ± 1.7 mm for females in the CT group, and 8.4 ± 2.7 mm for males and 7.5 ± 2.3 mm for females in the TKA group. After adjusting for the LAP dimension, there were significant gender differences in the ALCH in both the CT and TKA groups (P<0.01; Table 1).

The mean AMCH in males was 3.5 ± 1.9 mm in the CT groups, which is significantly larger than the mean in females (2.7 ± 1.6 mm; P<0.01). The mean AMCH was 4.4 ± 2.4 mm for males and 4.3 ± 2.6 mm for females in the TKA group, showing no significant gender difference (P = 0.418). After adjusting for the MAP dimension, there was a significant gender difference in the AMCH in both the CT and TKA groups (P<0.01; Table 2).

In addition, the dimensions of the anterior condylar height were highly variable, regardless of gender. The range of ALCH was 2.2–13.1 mm for males and 0.5–10.9 mm for females in the CT group, and 1–15.5 mm for males and 1.5–15.5 mm for females in the TKA group. The range for AMCH was 0–8.6 mm for males and 0–7.3 mm for females in the CT group, and 0–13 mm for males and 0–15 for females in the TKA group.

### Correlations between ALCH and the LAP dimension

There were significant negative correlations between the ALCH and LAP dimensions in both the CT and TKA groups, with the LAP dimension increasing as ALCH decreased regardless of gender. The line for males is above the line for females, indicating that males generally have a larger ALCH than females for a given MAP dimension (Fig 3).

### Correlations between AMCH and the MAP dimension

There were significant negative correlations between AMCH and the MAP dimension in both the CT and TKA groups, with the MAP dimension increasing as AMCH decreased regardless

**Table 2. Anterior medial condylar dimensions (mm).**

|  | Male | Female | P-value | Adjusted for MAP |
|---|---|---|---|---|
| CT group | 3.53 ± 1.94 | 2.66 ± 1.57 | <0.01 | <0.01 |
| TKA group | 4.44 ± 2.42 | 4.29± 2.56 | 0.418 | <0.01 |

Values are mean ± standard deviation. MAP: medial anterior-posterior

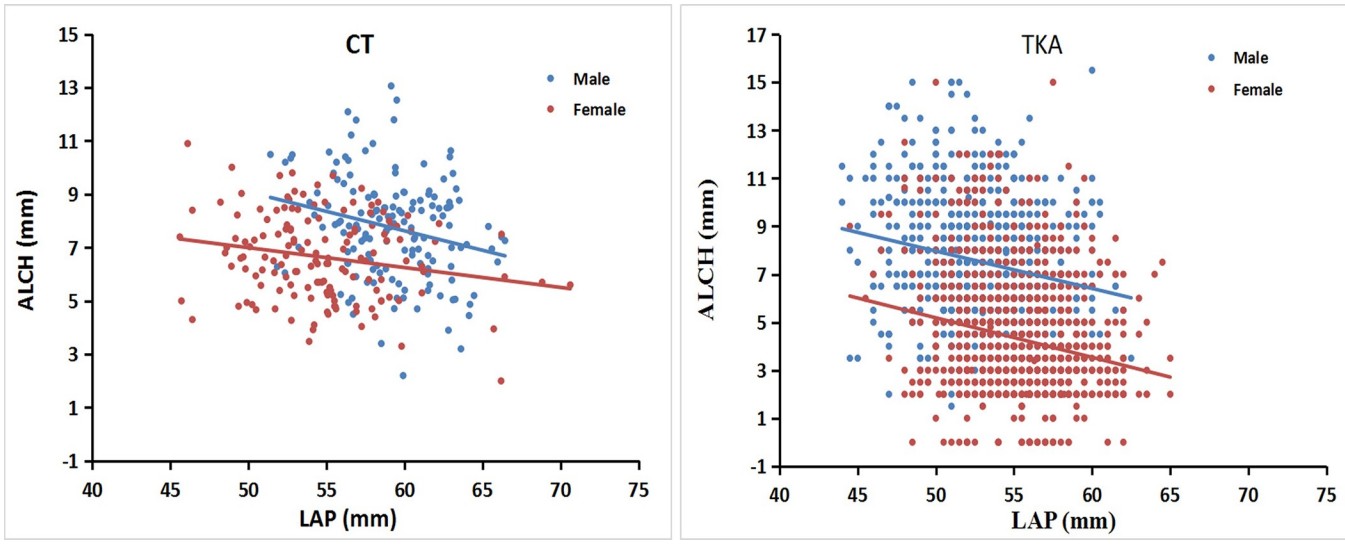

**Fig 3. The ALCH against the LAP dimensions in males and females.** ALCH decreased as the LAP measurement increased. (a) CT group. (b) TKA group.

of gender. The line for males is above the line for females, indicating that males generally have a larger AMCH than females for a given MAP dimension (Fig 4).

## Discussion

The most important finding in this study was that anterior femoral condyle heights decreased with increasing AP dimensions in both the medial and lateral condyle, which contradicts the current femoral component design in which the anterior flange height increases with increasing AP dimensions.

The anterior condylar height is one of the major anatomical differences taken into consideration when designing femoral components. Many studies have reported the anterior condylar anatomy using different methods. Gillespie et al measured 1207 skeletal cadaver femora

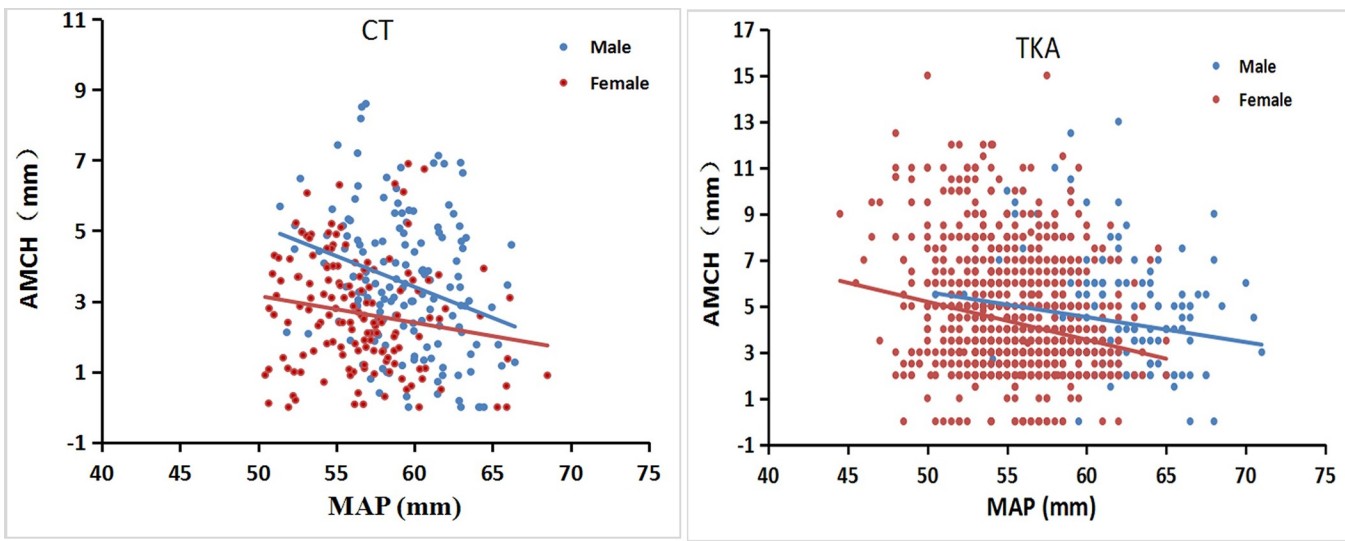

**Fig 4. The AMCH against the MAP dimensions in males and females.** AMCH decreased as the MAP measurement increased. (a) CT. (b) TKA.

(547 males and 660 females) and found that the anterior condyle height is related to the length of the femur, with a mean increase in the ALCH of 0.03 mm or AMCH of 0.02 mm for every 1 mm increase in the femoral length [11]. Fehring et al studied 212 knees in magnetic resonance (MR) images and reported that the average ALCH and AMCH was positive with medial-lateral (ML) size of the femur [6]. These values corroborate the findings of Li et al, who also reported that the thickness of the ALCH and AMCH was positive with ML size of the femur [10]. According to the above results, anterior flange heights in femoral components currently increase with increasing component size.

It has been widely accepted that restoration of native patellofemoral joint thickness at the time of TKA is important. Increasing patellar bone or anterior condyle thickness during TKA may cause patellofemoral joint overstuffing, which may lead to decreased flexion and possibly pain after TKA [9,13]. Bracey et al evaluated the effect of patellar thickness on knee flexion and patellar kinematics and found that knee flexion decreased an average 1.2˚ with each additional 2 mm of patellar thickness. In addition, kinematic tracking data showed a significantly greater lateral shift and tilt of the patella with the +8 mm prostheses [14]. Oishi et al examined varying amounts of patellofemoral build up in a cadaver model and reported that a 2 mm or 4 mm increase in patellar button thickness resulted in a significant increase in shear force at flexion angles greater than 40˚, which may lead to early failure of the implant due to loosening or increased wear [15]. In a cadaveric knee study, Mihalko et al reported a 4 mm anterior buildup on the femur that resulted in an approximately 4˚ loss of passive knee flexion [16]. Kawahara intraoperatively measured patellofemoral contact forces using three different femoral components in deep knee flexion and found that an increase in the AP dimensions of the femoral component significantly increased the patellofemoral contact forces at 120˚, 130˚, and 140˚ of flexion compared with the standard component [12].

Our data indicate that anterior femoral condyle height decreased with increasing AP dimensions in both the medial and lateral condyle (see Figs 3 and 4). One can infer from our data that the replacement of a knee with a small anterior condyle (large femur) with an implant with a thicker anterior flange can cause overstuffing, and the replacement of a knee with a large anterior condyle (small femur) with an implant with a thinner anterior flange can cause understuffing when using contemporary TKA components. Thus, to better fit the anterior femoral condyles, the small femoral implant should have a thicker anterior flange, and the large femoral implant should have a thinner anterior flange. Theoretically, the design of femoral components should more closely approximate the thickness of the original resected anterior condyle. Patellofemoral component design influences the patellofemoral kinematics [4, 17], which may ultimately contribute to some of the patellofemoral problems observed after TKA, such as pain or loss of motion and extensor mechanism deficiency [6, 18, 19]. Whether modification of the prosthetic anterior condyle would change the kinematics at the patellofemoral joint and offer a clinical advantage requires further study.

Most recent studies have focused on the gender differences in anterior condylar height. Poilvache et al measured the anterior condyle height of 46 males and 54 females after making the anterior cut and reported an average 1.4 mm lower ALCH and 1.6 mm lower AMCH for females compared with males [20]. Lonne et al intraoperatively studied the distal femurs of 100 males and 100 females and made similar observations; they reported an average difference between males and females of 1.48 mm anterolaterally and 0.76 mm anteromedially [21]. Other researchers recognized that gender differences were the result of a smaller femur in females. These data were direct, absolute measurements uncorrected for the size of the distal femur or patient height. Fehring et al measured the anterior condylar heights of males and females on MR images and found a significant difference in AMCH but no difference in

ALCH [6]. A recent study using 3D anatomic knee models reported that the difference between the sexes with regard to anterior condylar anatomy averaged 1.1 mm anterolaterally and 0.9 mm anteromedially [10]. However, the gender differences were no longer significant after adjusting for ML femoral size. Our data showed significant gender difference in ALCH and AMCH on CT, but only ALCH in the TKA group. In further analysis to eliminate the effect of femoral size, we found that the AMCH and ALCH of female knees were smaller than the size in corresponding male knees after adjusting for MAP and LAP measurements. We found significant gender differences in the lateral and medial anterior condylar heights in both the CT and TKA groups. Based on a large number of knees measured intraoperatively and on CT, we conclude that gender differences exist with respect to anterior condylar height. The discrepancy between the results from our study and other studies can be explained by anatomic variations among ethnicities and using different methods. Several studies have demonstrated ethnic differences in the shape and size of the knee [22, 23].

Our study has several advantages. First, the present study used a large sample size, giving it the power to verify significant features in anterior condylar anatomy. Second, the data in this study, measured in TKA patients after making the bone cut, accurately reflects the real clinical situation. Furthermore, via a virtual TKA procedure that simulated an anterior condyle cut on 3D CT knee models, we were able to obtain consistent results, strengthening the intraoperative measurement. A limitation was that all of the subjects recruited in the present study were from the Chinese population. Therefore, the subjects in the current study only represent a subgroup of the Chinese population, and other ethnic populations should be studied to verify our results.

## Conclusion

This study demonstrated that anterior femoral condylar height decreased with increasing AP dimensions in the medial and lateral condyle, which contradicts the present design in which the anterior flange increases with increasing AP dimensions. This study also showed that anterior condylar heights are highly variable, with gender differences. The data may provide an important reference for designing femoral anterior flange thickness to precisely match the natural trochlear anatomy.

## Supporting information

**S1 File. Support data.**
(ZIP)

## Author Contributions

**Conceptualization:** Xi Gong, Jia-Kuo Yu.

**Data curation:** Bo Yang.

**Funding acquisition:** Jia-Kuo Yu.

**Methodology:** Jia-Kuo Yu.

**Supervision:** Hai-jun Wang, Xi Gong.

**Writing – original draft:** Bo Yang.

**Writing – review & editing:** Fu-zhen Yuan, Yan-hai Chang, Jia-Kuo Yu.

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
