## [Decision Letter · Decision Letter 0]

26 May 2023

PONE-D-23-04989Femoral anterior condyle height decreased as the distal anteroposterior size increasing: contrary to the current designed femoral componentPLOS ONE

Dear Dr. yang,

Thank you for submitting your manuscript to PLOS ONE. After careful consideration, we feel that it has merit but does not fully meet PLOS ONE’s publication criteria as it currently stands. Therefore, we invite you to submit a revised version of the manuscript that addresses the points raised during the review process.

We look forward to receiving your revised manuscript.

Kind regards,

Gennaro Pipino, Md

Academic Editor

PLOS ONE

Journal Requirements:

"This research was funded by Instrument Research Project of the National Natural Science Foundation (No. 81327001). Science and Technology Program of Shaanxi Province (No.2022SF-049)."

6. We note that you have stated that you will provide repository information for your data at acceptance. Should your manuscript be accepted for publication, we will hold it until you provide the relevant accession numbers or DOIs necessary to access your data. If you wish to make changes to your Data Availability statement, please describe these changes in your cover letter and we will update your Data Availability statement to reflect the information you provide

7. PLOS requires an ORCID iD for the corresponding author in Editorial Manager on papers submitted after December 6th, 2016. Please ensure that you have an ORCID iD and that it is validated in Editorial Manager. To do this, go to ‘Update my Information’ (in the upper left-hand corner of the main menu), and click on the Fetch/Validate link next to the ORCID field. This will take you to the ORCID site and allow you to create a new iD or authenticate a pre-existing iD in Editorial Manager. Please see the following video for instructions on linking an ORCID iD to your Editorial Manager account: https://www.youtube.com/watch?v=_xcclfuvtxQ.

Additional Editor Comments:

As you know, the manuscript has a strong title and theme. Which sets the stage for a true revolution of the modern concept of knee prosthesis.

In general, as you can read from the following reports of the reviewers, it emerged that in order to suggest a new prosthetic design, more knowledge on many aspects is needed.

Reviewers' comments:

Reviewer's Responses to Questions

**Comments to the Author**

1. Is the manuscript technically sound, and do the data support the conclusions?

Reviewer #1: Yes

Reviewer #2: Partly

Reviewer #3: Partly

2. Has the statistical analysis been performed appropriately and rigorously? 

Reviewer #1: Yes

Reviewer #2: Yes

Reviewer #3: No

3. Have the authors made all data underlying the findings in their manuscript fully available?

Reviewer #1: Yes

Reviewer #2: Yes

Reviewer #3: Yes

4. Is the manuscript presented in an intelligible fashion and written in standard English?

Reviewer #1: Yes

Reviewer #2: Yes

Reviewer #3: No

5. Review Comments to the Author

Reviewer #1: 1. GENERAL COMMENTS

Manuscript topic is current and of interest.

The sample of patients completed the survey was satisfactory.

The English language is ok.

The study contains some anatomical features. Although there are some limitations, I would accept it for publication because it presents some interesting things related to total knee arthroplasty.

2. INTRODUCTION

The introduction provide the necessary background.

The purpose is clear.

3. METHODS (Material, patients and methods)

The design of the study is acceptable.

4. RESULTS

ok

4. DISCUSSION

The discussion is ok, although you should state that there are many designs that take into account the dimensions of the femur and that with the newest techniques like robotics the surgeon can calculate the amount of the bone that is necessary to be cut and adjust it to the specific implant design to be used in each patient.

6. CONCLUSION

ok

7. ABSTRACT

ok

8. TITLE

The title is ok.

9. REFERENCES

ok

10. TABLES

ok

Reviewer #2: Dear authors, i had the opportunity of reviewing your work, and i hope that my comments help strengthening your manuscript. You disclosed that you received no funds, however, later on you acknowledge your funding agency, please correct accordingly.

TITLE: please indicate in the title that this was a comparative study.

ABSTRACT: please made it clear that the measurements were clinical in the TKA cases and radiographic in the CT scans

INTRODUCTION:-references 10,11 in line 82 are old, please update

-lines 85 to 87: seems out of context, please correct or remove

-please rephrase the aim of the study and report it into primary objective and secondary objectives, as the authors through the methods and results reported on may differences not only the assessment of the anterior condylar height.

METHODS: - why the data was collected a long time ago (till 2015), it is almost 8 years, where major changes were made in the design of the implants, furthermore, as long as this was a historical group, why the authors did not report any data regarding the development of PFJ complications which is the main concerned point with their study.

-the authors reported using one implant for all of their cases, i was wondering why they did not report a comparison between the sizes of the femoral component anterior flange and compare it against their population. Furthermore, as long as they already know the femoral component sizes placed in each patient, they could have tested theses sizes against the measured resected anterior femoral cut.

-The authors did not report any inclusion or exclusion criteria, did they include all forms and magnitudes of deformities? patients with different pathologies such as RA, or patients with posttraumatic OA where the anatomy could be distorted?

-What was the exact technique for performing the TKA especially the rotation which could affect the anterior femoral condyles cut thickness?

RESULTS:

-the authors concentrated their comparisons on the gender differences, supposing that the CT group were a normal group, why the authors did not compare both groups to detect if there was a disturbance according to the pathological process leading to performing TKA.

-Could the differences between bone cuts thickness between males and females be attributed to the fact that the male skeleton is larger than the female skeleton?

-the authors stated that "the dimensions of the anterior condylar height were highly variable, regardless of

gender.", how could this be solved by changing the implants design? could a custom made implant for each patient becomes the solution?

DISCUSSION:

-Please indicate some possible solutions in view of your results.

Reviewer #3: Thank you for giving me the opportunity to review this manuscript. The authors describe differences in the dimensions regarding distal femur morphology of patients undergoing TKA and participants with "healthy" knees

Although I believe the manuscript would have some interest for the reader, I believe it is not ready for publication in the current form since there are significant methodological and language issues which cannot be ignored.

General comment:

I've never read or heard the term "condyle" regarding the anterior part of the distal femur. Usually the terms "anterior cortex" etc. are used.

Major issues:

There are many grammatical errors throughout the whole manuscript, which in some parts makes the reading/understanding of the text almost impossible. There is an urgent need for editing by a native speaker. Otherwise I do not see a chance for publication.

According to the findings of this study. There must be a high rate of patients with anterior overstuffing which should be visible in an impairment of clinical function or patient reported outcome measurements. It should be considered to include clinical data. Additionally, overstuffing should be measurable in postop. radiological evaluation. It should also be considered to include post-TKA CT and correlate the restoration of the anterior cortex by the anterior flange.

Methods:

Why do the authors compare healthy and degenerate knees? The implications of the study will mainly influence patients undergoing TKA. Additionally, patients undergoing ACL-reconstruction do not necessarily have a "healthy" knee anatomy.

The authors need to deliver more details on the intraoperative evaluations:

It is hard to define "flush" when it comes to the anterior cut of the distal femur. What was the exact definition in your study?

Were varus and valgus deformity included?

How much external rotation was used for the distal femoral cut?

Results:

Can the authors explain the different results from CT and intraop.?

Line 180 and following: I cannot find the calculations for the correlations between ALCH and LAP and AMCH and MAP respectively. Correlations should be calculated (e.g. Spearman) to display whether these findings show a significance. The method should then also be named in the M+M section.

Discussion:

Line 197-206: The authors report on studies which show results in contradiction to their findings and which support the rationale of contemporary TKA designs  the bigger the femoral component size the bigger the anterior flange. However, the authors do not elucidate why these differences in their study are explainable.

6. PLOS authors have the option to publish the peer review history of their article (what does this mean?). If published, this will include your full peer review and any attached files.

Reviewer #1: No

Reviewer #2: No

Reviewer #3: No

---

## [Author Response · Author response to Decision Letter 0]

24 Aug 2023

Dear Editors and Reviewers:

Thank you very much for your letter and for the reviewers’ careful reading and constructive comments concerning our manuscript entitled “Femoral anterior condyle height decreased as the distal anteroposterior size increasing: contrary to the current designed femoral component” (Manuscript # PONE-D-23-04989). According to editor and reviewers' comments and suggestions, we have revised the manuscript point by point. 

We would like to resubmit the revised manuscript to PLOS ONE, and hope it is acceptable for publication in the journal. The detailed changes were tracked in red in the revised manuscript and listed in the following:

Looking forward to hearing from you soon. 

Best wishes!

For editors:

1. Please ensure that your manuscript meets PLOS ONE's style requirements, including those for file naming. The PLOS ONE style templates can be found at https://journals.plos.org/plosone/s/file?id=wjVg/PLOSOne_formatting_ sample_main_body.pdf and https://journals.plos.org/plosone/s/file?id=ba62 /PLOS One_formatting_sample_title_authors_affiliations.pdf

Answer: Thank you for your good comments. We have edited the manuscript according to PLOS ONE's style requirements.

"This research was funded by Instrument Research Project of the National Natural Science Foundation (No. 81327001). Science and Technology Program of Shaanxi Province (No.2022SF-049)."

Please remove any funding-related text from the manuscript and let us know how you would like to update your Funding Statement. Currently, your Funding Statement reads as follows: "The author(s) received no specific funding for this work."

Answer: Thank you for your careful reading and suggestions, we have changed funding-related text in revised manuscript and cover letter.

Answer: Thank you very much, we have corrected grant numbers in the ‘Funding Information’ section.

Answer: Thank you, we have changed funding-related text in revised manuscript and cover letter.

Answer: Thank you very much, we have provide the related figures and table in supporting information. But we can not provide detailed information for our data. Because we are applying for a patent and designing mew femoral prosthesis according to the data now. thank you.

6. We note that you have stated that you will provide repository information for your data at acceptance. Should your manuscript be accepted for publication, we will hold it until you provide the relevant accession numbers or DOIs necessary to access your data. If you wish to make changes to your Data Availability statement, please describe these changes in your cover letter and we will update your Data Availability statement to reflect the information you provide

Answer: Thank you for your kindly remind us that we can not provide repository information for our data after acceptance. Because of we are applying for a patent and designing mew femoral prosthesis according to the data now. we have described it online and in the cover letter.

7. PLOS requires an ORCID iD for the corresponding author in Editorial Manager on papers submitted after December 6th, 2016. Please ensure that you have an ORCID iD and that it is validated in Editorial Manager. To do this, go to ‘Update my Information’ (in the upper left-hand corner of the main menu), and click on the Fetch/Validate link next to the ORCID field. This will take you to the ORCID site and allow you to create a new iD or authenticate a pre-existing iD in Editorial Manager. Please see the following video for instructions on linking an ORCID iD to your Editorial Manager account: https://www.youtube.com/watch?v=_xcclfuvtxQ.

Answer: Thank you，I have registered a new ORCID iD.

Answer: Thank you very much. 

Additional Editor Comments:

As you know, the manuscript has a strong title and theme. Which sets the stage for a true revolution of the modern concept of knee prosthesis.

In general, as you can read from the following reports of the reviewers, it emerged that in order to suggest a new prosthetic design, more knowledge on many aspects is needed.

Answer: Thank you very much. We have revised the manuscript point by point as fellows according to reviewers’ comments.

Reviewers' comments:

Reviewer's Responses to Questions

Comments to the Author

1. Is the manuscript technically sound, and do the data support the conclusions?

Reviewer #1: Yes

Reviewer #2: Partly

Reviewer #3: Partly

Answer: Thank you for your comments. We have amended the whole manuscript carefully in the revised manuscript.

2. Has the statistical analysis been performed appropriately and rigorously?

Reviewer #1: Yes

Reviewer #2: Yes

Reviewer #3: No

Answer: Thank you for your comments. We have amended the problems of statistical analysis in the revised manuscript.

3. Have the authors made all data underlying the findings in their manuscript fully available?

Reviewer #1: Yes

Reviewer #2: Yes

Reviewer #3: Yes

Answer: Thank you very much.

4. Is the manuscript presented in an intelligible fashion and written in standard English?

Reviewer #1: Yes

Reviewer #2: Yes

Reviewer #3: No

Answer: Thank you very much. Our manuscript had been edited for English-language problems by Paul Kretchmer, Managing Director, San Francisco Edit.

5. Review Comments to the Author

Answer: Thanks. We have answered the questions above.

Reviewer #1: 

1.GENERAL COMMENTS

Manuscript topic is current and of interest.

The sample of patients completed the survey was satisfactory.

The English language is ok.

The study contains some anatomical features. Although there are some limitations, I would accept it for publication because it presents some interesting things related to total knee arthroplasty.

Answer: Thank you very much for your good praise.

2. INTRODUCTION

The introduction provide the necessary background.

The purpose is clear. 

Answer: Thank you for your comments.

3. METHODS (Material, patients and methods)

The design of the study is acceptable.

Answer: Thank you.

4. RESULTS

ok 

Answer: Thanks.

5. DISCUSSION

The discussion is ok, although you should state that there are many designs that take into account the dimensions of the femur and that with the newest techniques like robotics the surgeon can calculate the amount of the bone that is necessary to be cut and adjust it to the specific implant design to be used in each patient.

Answer: Thank you for your comments.

6. CONCLUSION

ok 

Answer: Thanks again.

7. ABSTRACT

ok

Answer: Thank you.

8. TITLE

The title is ok.

Answer: Thanks.

9. REFERENCES

ok

Answer: Thank you very much.

10. TABLES

ok

Answer: Thank you again.

Reviewer #2: 

Dear authors, i had the opportunity of reviewing your work, and i hope that my comments help strengthening your manuscript. You disclosed that you received no funds, however, later on you acknowledge your funding agency, please correct accordingly.

Answer: Thank you for your careful reading and suggestions, we have changed funding-related text in revised files.

TITLE: please indicate in the title that this was a comparative study.

Answer: Thank you. We have revised the title as “Femoral anterior condyle height decreases as the distal anteroposterior size increases in total knee arthroplasty: A comparative study ” in revised manuscript(line 5-6).

ABSTRACT: please made it clear that the measurements were clinical in the TKA cases and radiographic in the CT scans.

Answer: Thank you, we have described the measurements were undergoing TKA intraoperative and computed tomography scans in revised manuscript (line 45-46).

INTRODUCTION:

-references 10,11 in line 82 are old, please update

Answer: Thank you for your suggestion, we have updated the references 10,11 in revised manuscript(line 297-299).

10. Li P, Tsai TY, Li JS, Wang S, Zhang Y, Kwon YM, et al. Gender analysis of the anterior femoral condyle geometry of the knee. Knee. 2014; 21(2): 529-533. 

11. Gillespie RJ, Levine A, Fitzgerald SJ, Kolaczko J, DeMaio M, Marcus RE, et al. Gender differences in the anatomy of the distal femur. J Bone Joint Surg Br. 2011; 93(3):357-63. 

lines 85 to 87: seems out of context, please correct or remove

Answer: Thank you for your suggestion, we have corrected these sentences in the revised manuscript(line 82-85).

please rephrase the aim of the study and report it into primary objective and secondary objectives, as the authors through the methods and results reported on may differences not only the assessment of the anterior condylar height.

Answer: Thank you for your good suggestion. We have added secondary objectives that whether gender differences exist in anterior condyle height in the revised manuscript(line43-44).

METHODS: - why the data was collected a long time ago (till 2015), it is almost 8 years, where major changes were made in the design of the implants, furthermore, as long as this was a historical group, why the authors did not report any data regarding the development of PFJ complications which is the main concerned point with their study.

Answer: This study mainly clarified the anatomy features we found that the femoral anterior condyle height decreases with the increase of the distal femur dimension, which is different from the prosthesis currently used (femoral anterior flange increases with the prosthesis size increased). During that period, we didn't publish a paper because we applied for a patent and designed mew femoral prosthesis according to the data. In clinic, some patients have PFJ-related complications after TKA, including prosthesis overfilling, which is partly due to that the anterior flange thickness of prosthesis is greater than that of anterior condyle resection thickness. I hope I can answer your question. Thank you.

-the authors reported using one implant for all of their cases, i was wondering why they did not report a comparison between the sizes of the femoral component anterior flange and compare it against their population. Furthermore, as long as they already know the femoral component sizes placed in each patient, they could have tested theses sizes against the measured resected anterior femoral cut.

Answer: Thanks, the question you raised is very good, and we have neglected this aspect. Later, we will collect relevant data and make a summary. This paper mainly wants to explain the anatomy features we found that the femoral anterior condyle height decreases with the increase of the distal femur dimension, which is different from the prosthesis currently used (femoral anterior flange increases with the prosthesis size increased).

-The authors did not report any inclusion or exclusion criteria, did they include all forms and magnitudes of deformities? patients with different pathologies such as RA, or patients with posttraumatic OA where the anatomy could be distorted?

Answer: Thank you. The criteria for inclusion and exclusion of cases are very important. We choose primary OA subjects for this study. Patients were excluded if they had a history of femur fracture or congenital anomaly or if the bone loss or degradation was so serious that augmentation was required or if the knee had a varus or valgus deformity of >15°. We have described it in the revised manuscript(line 95-99).

What was the exact technique for performing the TKA especially the rotation which could affect the anterior femoral condyles cut thickness? 

Answer: Generally, The femoral external rotation 3°relative to the posterior condylar axis for the distal femoral cut, or paralleling the transepicondylar axis or perpendicular to Whitesides’ line to ensure the flexion gap balance. We have described it in the revised manuscript(line 102-104). 

RESULTS:

The authors concentrated their comparisons on the gender differences, supposing that the CT group were a normal group, why the authors did not compare both groups to detect if there was a disturbance according to the pathological process leading to performing TKA.

Answer: Thank you, it is a good question. The pathology of TKA patients may have some influences on the anatomy of knee joint. The cartilage of TKA patients was partly worn or even disappeared completely, and there was no cartilage exit in CT image. In addition, the present study mainly to elucidate the anatomical features that anterior condyle height decreased as the increase of the distal femur. CT images and TKA intraoperative measurements verify the results for each other.

Could the differences between bone cuts thickness between males and females be attributed to the fact that the male skeleton is larger than the female skeleton?

Answer: Many literatures have previously reported that gender differences of knee anatomy were the result of a smaller skeleton in females. Femoral anterior condylar cuts thickness between males and females partly attributed to the fact that the male skeleton is larger than the female skeleton. In addition, we found significant gender differences of anterior condylar resection thickness. 

The authors stated that "the dimensions of the anterior condylar height were highly variable, regardless of gender.", how could this be solved by changing the implants design? could a custom made implant for each patient becomes the solution?

Answer: Our results showed that there is a negative correlation between the femoral anterior condyle height and the distal femur size. But the individual is highly variable. Custom-made prostheses are the best solution, and many researchers and companies are now working on personalized design and 3D printing. our research group also has been working in this area, and we believe it can be realized in the near future.

DISCUSSION:

-Please indicate some possible solutions in view of your results.

Answer: Thanks, according to our results, in order to better fit the femoral anterior condyles, when designed femoral prosthesis, the large femoral implant should have a thinner anterior flange, and the small femoral implant should have a thicker anterior flange. Theoretically, femoral components designed like this should more closely approximate the thickness of the original resected anterior condyle. 

Reviewer #3: Thank you for giving me the opportunity to review this manuscript. The authors describe differences in the dimensions regarding distal femur morphology of patients undergoing TKA and participants with "healthy" knees.

Although I believe the manuscript would have some interest for the reader, I believe it is not ready for publication in the current form since there are significant methodological and language issues which cannot be ignored.

Answer: Thank you very much. We have edited the language issues of whole manuscript by a native speaker.

General comment:

I've never read or heard the term "condyle" regarding the anterior part of the distal femur. Usually the terms "anterior cortex" etc. are used.

Answer: Thank you for your question. The term "condyle" regarding the anterior part of the distal femur usually used the terms "anterior cortex" etc. But some authors use it too, such as: 

Fehring TK, Odum SM, Hughes J, Springer BD, Beaver WB Jr. Differences between the sexes in the anatomy of the anterior condyle of the knee. J Bone Joint Surg Am. 2009; 91-A: 2335-2341.

Ishitani K, Ito T, Hatada R, Kuriyama S, Nakamura S, Ito H, Matsuda S. High and varied anterior condyle of the distal femur is associated with limited flexion in varus knee osteoarthritis. Cartilage. 2021;13(1suppl):1487S-1493S.

If you don't think it's appropriate, later we will change it. Thanks again.

Major issues:

There are many grammatical errors throughout the whole manuscript, which in some parts makes the reading/understanding of the text almost impossible. There is an urgent need for editing by a native speaker. Otherwise I do not see a chance for publication.

Answer: Thank you for your comments. Our manuscript had been edited for English-language problems by a native speaker, Paul Kretchmer, Managing Director, San Francisco Edit.

According to the findings of this study. There must be a high rate of patients with anterior overstuffing which should be visible in an impairment of clinical function or patient reported outcome measurements. It should be considered to include clinical data. Additionally, overstuffing should be measurable in postop. radiological evaluation. It should also be considered to include post-TKA CT and correlate the restoration of the anterior cortex by the anterior flange. 

Answer: Thanks, the question you raised is very good. According to our results, there must be a high rate of patients with anterior overstuffing in bigger knee or understuffing in smaller knee, which finally results in function impairment in clinical, such as anterior knee pain, limiting postoperative knee range of motion, etc. These could be evaluated by radiological method after TKA. However, the mainly purpose in this paper was to clarity the anatomy features that femoral anterior condyle height decreases as the femoral AP size increases. The clinical and radiological evaluation requires further study. Later, we will collect relevant clinical and radiological data, and make a summary. I hope I can answer your question. Thank you again.

Methods: Why do the authors compare healthy and degenerate knees? The implications of the study will mainly influence patients undergoing TKA. Additionally, patients undergoing ACL-reconstruction do not necessarily have a "healthy" knee anatomy.

Answer: Thank you very much, it is a good question. The present study mainly to elucidate the anatomical features of femoral anterior condyle height decreased as femoral AP size increases. CT images and TKA intraoperative measurements verify the results for each other. In addition, we selected ACL reconstruction patients with normal knee anatomy, we have described it in revised manuscript(line 123-125). 

The authors need to deliver more details on the intraoperative evaluations: It is hard to define "flush" when it comes to the anterior cut of the distal femur. What was the exact definition in your study? 

Answer: Thank you very much. Anterior referencing instrumentation was used to cut the anterior condyle. The anterior condylar heights were measured intraoperatively only for patients whose anterior condylar cut line flushed with the anterior cortical surface of the distal femoral shaft. We Illustrate it with a legend (fig 1a) in revised manuscript (line 102-107). 

Answer: 

Were varus and valgus deformity included?

Answer: Thank you. The criteria for inclusion and exclusion of cases are very important. Patients were excluded if they had a history of femur fracture or congenital anomaly or if the bone loss or degradation was so serious that augmentation was required or if the knee had a varus or valgus deformity of >15°. We have described it in the revised manuscript (line 95-99). 

How much external rotation was used for the distal femoral cut?

Answer: Thank you. Generally, The femoral external rotation 3°relative to the posterior condylar axis for the distal femoral cut, or paralleling the transepicondylar axis or perpendicular to Whitesides’ line to ensure the flexion gap balance. We have described it in the revised manuscript (line 102-105). 

 Results:

Can the authors explain the different results from CT and intraop.?

Answer: Thank you, it is a good question. The pathology of TKA patients have some influences on the anatomy of knee. The cartilage thickness of TKA cases was partly worn or even disappeared completely, and there was no cartilage exit in CT image. So we didn't compare their differences. In addition, the present study mainly to elucidate the anatomical features of femoral anterior condyle height decreased as femoral AP size increases. CT images and TKA intraoperative measurements verify the results for each other.

Line 180 and following: I cannot find the calculations for the correlations between ALCH and LAP and AMCH and MAP respectively. Correlations should be calculated (e.g. Spearman) to display whether these findings show a significance. The method should then also be named in the M+M section.

Answer: Thank you, we apply linear regression analysis to determine correlations between ALCH and LAP, AMCH and MAP dimensions. We have described it in the revised manuscript (line 154-155). 

Discussion:

Line 197-206: The authors report on studies which show results in contradiction to their findings and which support the rationale of contemporary TKA designs  the bigger the femoral component size the bigger the anterior flange. However, the authors do not elucidate why these differences in their study are explainable.

Answer:

Some studies have reported that the anterior condylar height have a positive correlation with the length of the femur or femoral ML size. There has not been a study that analyzed the relationship between anterior condyle height and distal femoral AP size. Our study found that the femoral anterior condyle height decreased with the distal femur size increase, which contrary to the prosthesis currently used. This is the anatomical feature of femoral anterior condyle that we found. 

6. PLOS authors have the option to publish the peer review history of their article (what does this mean?). If published, this will include your full peer review and any attached files. Do you want your identity to be public for this peer review? For information about this choice, including consent withdrawal, please see our Privacy Policy.

Reviewer #1: No

Reviewer #2: No

Reviewer #3: No

Answer: Thank you.

 Answer: Thank you we have deal the figures with PACE digital diagnostic tool and uploaded them to the system again.

---

## [Decision Letter · Decision Letter 1]

10 Jan 2024

Femoral anterior condyle height decreases as the distal anteroposterior size increases in total knee arthroplasty: A comparative study

PONE-D-23-04989R1

Dear Dr. yang,

We’re pleased to inform you that your manuscript has been judged scientifically suitable for publication and will be formally accepted for publication once it meets all outstanding technical requirements.

Kind regards,

Gennaro Pipino, Md

Academic Editor

PLOS ONE

Additional Editor Comments (optional):

Thank you for sending us your work. This is a clearly written manuscript on a interesting topic.

I'm sorry for the long times, but there were some mixed reviews among reviewers. But in the end I'm happy to tell you that your work has been accepted!

Reviewers' comments:

Reviewer's Responses to Questions

**Comments to the Author**

1. If the authors have adequately addressed your comments raised in a previous round of review and you feel that this manuscript is now acceptable for publication, you may indicate that here to bypass the “Comments to the Author” section, enter your conflict of interest statement in the “Confidential to Editor” section, and submit your "Accept" recommendation.

Reviewer #1: All comments have been addressed

Reviewer #2: (No Response)

Reviewer #4: All comments have been addressed

2. Is the manuscript technically sound, and do the data support the conclusions?

Reviewer #1: Yes

Reviewer #2: Partly

Reviewer #4: Yes

3. Has the statistical analysis been performed appropriately and rigorously? 

Reviewer #1: Yes

Reviewer #2: I Don't Know

Reviewer #4: Yes

4. Have the authors made all data underlying the findings in their manuscript fully available?

Reviewer #1: Yes

Reviewer #2: No

Reviewer #4: Yes

5. Is the manuscript presented in an intelligible fashion and written in standard English?

Reviewer #1: Yes

Reviewer #2: Yes

Reviewer #4: Yes

6. Review Comments to the Author

Reviewer #1: (No Response)

Reviewer #2: Dear authors, thanks for submitting your revised manuscript. Unfortunately, i found it still not appropriate for publication. A lot of data should be explained and added to the study to make it more valuable. Thanks for the effort.

Reviewer #4: Having read the authors' responses to the reviewers, I think the impact of this work is interesting.

In conclusion, the study certainly has interesting points because it deals with a topic of great interest. The purpose is clear and respected. The main question addressed by the research is clear and completely agreeable. I believe that the information provided is sufficient and represents useful elements to encourage the development of new scientific work.

7. PLOS authors have the option to publish the peer review history of their article (what does this mean?). If published, this will include your full peer review and any attached files.

Reviewer #1: **Yes: **KONSTANTINOS G. MAKRIDIS

Reviewer #2: No

Reviewer #4: No

---

## [Editor Report · Acceptance letter]

16 Feb 2024

PONE-D-23-04989R1 

PLOS ONE

Dear Dr. Yang, 

I'm pleased to inform you that your manuscript has been deemed suitable for publication in PLOS ONE. Congratulations! Your manuscript is now being handed over to our production team.

Kind regards, 

on behalf of

Professor Gennaro Pipino 

Academic Editor

PLOS ONE